# Accumulation Mechanism and Risk Assessment of *Artemisia selengensis* Seedling In Vitro with the Hydroponic Culture under Cadmium Pressure

**DOI:** 10.3390/ijerph19031183

**Published:** 2022-01-21

**Authors:** Tao Tang, Wei Kang, Mi Shen, Lin Chen, Xude Zhao, Yongkui Wang, Shunwen Xu, Anhuai Ming, Tao Feng, Haiyan Deng, Shuqi Zheng

**Affiliations:** 1School of Resources and Environmental Engineering, Wuhan University of Science and Technology, Wuhan 430080, China; tangtaowkd@163.com (T.T.); Chenlin1735@163.com (L.C.); fengtaowhu@163.com (T.F.); 2Hubei Provincial Key Laboratory of Mining Area Environmental Pollution Control and Remediation, Hubei Polytechnic University, Huangshi 435003, China; 223004@hbpu.edu.cn (M.S.); zhaoxude@hbpu.edu.cn (X.Z.); 204055@hbpu.edu.cn (Y.W.); bx_ollie08051005@163.com (H.D.); Zhengshuqi2021@163.com (S.Z.); 3College of Environmental Engineering, Wuhan Textile University, Wuhan 430200, China; 4Huangshi Vegetable Industry Development Center, Huangshi 435003, China; xunshunwen2021@163.com (S.X.); minganhuai@163.com (A.M.)

**Keywords:** *Artemisia selengensis*, cadmium, subcellular distribution, cell wall, health risk

## Abstract

*Artemisia selengensis* is a perennial herb of the *Compositae* with therapeutic and economic value in China. The cadmium (Cd) accumulation mechanism and healthy risk evaluation of *A. selengensis* were investigated in this study. Tissue culture seedlings were obtained by plant tissue culture in vitro, and the effect of Cd stress (Cd concentration of 0.5, 1, 5, 10, 25, 50 and 100 μM) on *A. selengensis* was studied under hydroponic conditions. The results showed that low-Cd (0.5–1 μM) stress caused a rare effect on the growth of *A. selengensis* seedlings, which regularly grew below the 10 μM Cd treatment concentration. The biomass growth rate of the 0.5, 1, and 5 μM treatment groups reached 105.8%, 96.6%, and 84.8% after 40 days of cultivation, respectively. In addition, when the concentration of Cd was greater than 10 μM, the plant growth was obviously inhibited, i.e., chlorosis of leaves, blackening roots, destroyed cell ultrastructure, and increased malondialdehyde (MDA) content. The root could be the main location of metal uptake, 57.8–70.8% of the Cd was concentrated in the root after 40 days of cultivation. Furthermore, the root cell wall was involved in the fixation of 49–71% Cd by subcellular extraction, and the involvement of the participating functional groups of the cell wall, such as -COOH, -OH, and -NH_2_, in metal uptake was assessed by FTIR analysis. Target hazard quotient (THQ) was used to assess the health risk of *A. selengensis*, and it was found that the edible part had no health risk only under low-Cd stress (0.5 to 1 μM) and short-term treatment (less than 20 days).

## 1. Introduction

Cadmium (Cd) is one of the most toxic trace metals and is harmful to humans, animals and plants. The accelerated development of industry and agriculture has increased the content of Cd in the soil [1]. Agricultural soils in countries including Japan, China, Bangladesh, India and Australia are polluted by Cd to varying degrees [2,3,4]. Cd has higher bioavailability in soil [5] and mobility in plants [6], which makes it easy to transport from roots to above-ground parts compared with other trace metals [7]. Plants will have obvious symptoms of poisoning after accumulating Cd, such as growth inhibition, chlorosis, the browning of roots, or the death of the whole plant [8]. Due to its high bioavailability in the soil, the accumulation of Cd in crops grown on Cd-contaminated soil poses a serious threat to human health [5]. Crops contaminated with Cd are consumed by humans, which is considered to be the primary reason for Cd entry into humans [9]. The toxic properties of Cd act on many organs of the human body, but it mainly accumulates in the kidneys, causing serious damage, including emphysema, renal tubular damage and kidney stones [10]. Therefore, it is essential to guarantee good crop growth and food security when agricultural activities are carried out in Cd contaminated soil [11]. Solving the problem of soil Cd pollution and Cd accumulation in crops is urgent.

Various methods have been used to repair soil contaminated by heavy metals, such as soil solidification/stabilization, vitrification, the translocation of contaminated soil, physical stabilization, or heavy metal chelating agent cleaning of contaminated soil [12]. Immobilizers induce metal co-precipitation, adsorption and ion exchange, or promote humification and redox conversion, thereby reducing the solubility and bioavailability of Cd [13]. However, these technologies have disadvantages such as high cost, slow processing, and damage to the soil structure and ecological balance in practice. In addition, the demand for food production in developing countries is extremely high, and it is impossible for farmers to fallow farmland for remediation [12]. Therefore, screening and cultivating crop varieties with low-Cd accumulation is the most effective way to reduce the human absorption of Cd from crops [14]. The ability of crops to absorb and accumulate Cd varies significantly between species and populations [15]. In recent years, some low-Cd accumulation crops have been delimited. Duan et al. [16] screened six low-Cd accumulation rice varieties in soil conditions with Cd content of 0.4–0.5 mg/kg and pH 5.19–5.64. Nine of the 72 wheat cultivars were identified as low-Cd wheat and moderately high micronutrient concentrations in grain [17]. The exploration of the food safety of screening crops has reduced soil cadmium entering the human body through the food chain and eliminated the need for ploughing the land [18].

*Artemisia selengensis* is a perennial herb in the *Compositae* family, which is widely distributed along low-lying damp area, marshes, and lakeshores. Due to its medicinal and food value, for thousands of years in China, it has been favored as medicine—which is attributed to its anti-oxidant and anti-cancer activities [19,20]—and food to yield some functional products, such as beverages, tea, and yoghurt. Xu et al. [21] found that *A. selengensis* had a higher content of heavy metals in its leaves than stems and roots. However, only a few reports have focused on the accumulation characteristic of metals in the aqueous culture of the plant and the health risks associated with it. Given the above concerns above, the aim of this study was to investigate the accumulation characteristics of the *A. selengensis* seedling under Cd pressure to evaluate the potential risk. In this experiment, *A. selengensis* seedlings grown in wetlands of lakeshore in the wild were sampled; then, sterile tissue culture seedlings were obtained in vitro for the subsequent treatment. The seedlings were cultured under Cd gradient processing in MS aqueous solution, and the biomass and metal concentrations of plants were measured to understand the accumulation characteristics. The malondialdehyde (MDA) content was determined, and the intracellular morphology was observed with metal pressure to identify the tolerance of the seedlings. Furthermore, the Cd distribution was revealed by the separation of subcellular components and displaying the characteristic functional groups. Finally, the THQ (target hazard quotient) was used to assess the human health risks for consuming *A. selengensis,* which could provide a basis for the safe consumption of *A. selengensis*.

## 2. Materials and Methods

### 2.1. Plant Culture In Vitro and Cd Treatment

This experiment used wild *A. selengensis* detoxified tissue culture seedlings, cultivated in a plant culture room, with daytime conditions set at 12 h, 3000 lx, 65% humidity, and 25 °C. The seedlings of *A. selengensis* with similar growth were adapted for 7 days in 1/2 MS nutrient solution. The nutrient solution composition was as follows: NH_4_NO_3_ (33.0 g/L), KNO_3_ (38.0 g/L), KH_2_PO_4_ (3.40 g/L), MgSO_4_ (3.62 g/L), CaCl_2_ (6.65 g/L), KI (0.166 g/L), MnSO_4_ (3.02 g/L), H_3_BO_3_ (1.24 g/L), ZnSO_4_ (0.97 g/L), Na_2_MoO_4_·2H_2_0 (0.005 g/L), CoCl_2_·6H_2_0 (0.005 g/L), CuSO_4_·5H_2_0 (0.050 g/L). Then, *A. selengensis* seedlings with the same growing vigor were selected and transferred to 1/2 MS medium with different concentrations of CdSO_4_ (0.5, 1, 5, 10, 25, 50, and 100 μM), taking no CdSO_4_ as CK and shading the culture medium. The initial fresh weight and plant height of the whole plants were recorded, and each treatment had three replicates. Plants were harvested from the 10th, 20th, 30th, and 40th day of culture.

### 2.2. Analysis of Plant Biomass and Cd Content

The roots of the harvested plants were rinsed with deionized water 3 times to remove the Cd ions attached to the roots. Plant height was measured by vernier calipers and divided into three parts: root, stem, and leaf. Then, the fresh weight was recorded before 105 °C for 30 min and drying at 70 °C to constant weight. An amount of 0.5 g of dried plant sample was placed into a crucible containing 3 mL of concentrated nitric acid for heating and digestion. The sample after digestion was fixed to a volume of 0.2% (*v*/*v*) nitric acid solution as the test solution. An inductively coupled plasma optical emission spectrometer (ICP-OES, Thermo 7000 Waltham, MA, USA) was used to measure the Cd content. Each treatment was repeated three times.

### 2.3. Determination of Malondialdehyde (MDA) Content in Plants

An amount of 0.5 g of fresh sample was ground together with 2 mL of 10% (*w*/*v*) trichloroacetic acid (TCA) extract, and then 4 mL of TCA was added to further grind it to a homogenate. The homogenate was centrifuged at 4000 r/min for 10 min. Furthermore, 2 mL of the supernatant was mixed well with 2 mL 0.6% (*w*/*v*) thiobarbituric acid (TBA) solution. This mixture was heated in boiling water for 15 min, followed by quick cooling in running water. After centrifugation at 4000 r/min for 10 min, the supernatant was taken to measure the absorbance at 532, 600, and 450 nm, which were recorded as A_532_, A_600_ and A_450_, respectively. The concentration of MDA (μmol/L) = 6.45 × (A_532_-A_600_)-0.56 × A_450_. The experiment was repeated three times.

### 2.4. Extraction of Subcellular Components of Root

Fresh plant roots of 1.0 g were cut into small pieces and homogenized with the pre-cooling extraction buffer (0.25 M sucrose, 50 mM Tris-HCl (pH = 7.5), and 1 mM dithioerythritol) at a ratio of 1:10. All steps were performed at 4 °C. The resulting homogenate was centrifuged at 4000 r/min for 10 min. The lower sediment and bottom layer debris were the cell wall fraction (f1). Then, the supernatant was centrifuged at 12,000 r/min for 30 min. The lower sediment was the organelle fraction (f2), and the resultant supernatant was labeled as the soluble fraction (f3). All fractions were dried at 70 °C to a constant weight, and the Cd content was determined after digestion. The experiment was repeated three times.

### 2.5. Analysis of Transmission Electron Microscopy (TEM)

The plant tissues were embedded in 5% (*v*/*v*) glutaraldehyde solution and fixed overnight at 4 °C; the samples were rinsed 4 times with 0.1 mol/L phosphate buffer (pH = 7.0) for 15 min each time; the samples were then fixed with 1% (*v*/*v*) osmic acid solution for 1–2 h; the samples were rinsed 4 times with 0.1 mol/L phosphate buffer (pH = 7.0) for 15 min each time; after that, the sample was dehydrated step by step with ethanol solution (50%, 70%, 80%, 90%, and 95%), each concentration was treated for 15 min, and then 100% ethanol was used for one time for 20 min, and the sample was fixed. After sectioning, it was stained with saturated uranyl acetate aqueous solution for 25 min and lead citrate for 10 min. The cell morphology was observed using a transmission electron microscope (TEM, FEI Tecani G^2^0 F20 S-TWIN, FEI, Hillsboro, OR, USA).

### 2.6. Fourier Transform Infrared (FTIR) Spectroscopy

The root cell wall fractions were taken from the step described in Chapter 2.4. Then, the fraction was fully ground in an agate mortar at a weight ratio of 1:150 with KBr and compressed into a thin tablet. The tablet was scanned with a Fourier transform infrared spectroscopy analyzer (FTIR, Tensor27, Bruker, Billerica, MA, USA). The spectral resolution was 4 cm^−1^, and the infrared spectrum signal of the sample was recorded in the range of 4000–500 cm^−1^.

### 2.7. Statistical Analysis

The fresh weight (FW) rate was calculated as follows:$$\mathrm{FW}\;\mathrm{rate}\;\left(\%\right)=\frac{M}{M_{0}}\times 100\%$$
where M is the fresh weight of the whole plant at harvest; M_0_ is the initial fresh weight of whole plant before treatment with Cd stress.

The translocation factor (TF) can reflect the plant’s ability to accumulate Cd and the higher the TF is, the stronger the plant’s ability to transport heavy metals, as calculated in the following:$$\mathrm{TF}=\frac{\mathrm{Cd}\;\mathrm{content}\;\mathrm{of}\;\mathrm{the}\;\mathrm{above}-\mathrm{ground}\;\mathrm{part}}{\mathrm{Cd}\;\mathrm{content}\;\mathrm{of}\;\mathrm{the}\;\mathrm{root}}$$

The THQ (target hazard quotient) is used to assess the possibility of health risks in pollutants, and can be used to identify the non-carcinogenic risk of heavy metals to human health [22]. THQ < 1 indicates that heavy metals have no significant impact on human health. The higher the THQ value, the higher the possibility of long-term carcinogenic effects, as calculated in the following [23]:$$\mathrm{THQ}=\frac{E_{F}E_{D}F_{\mathrm{IR}}C}{R_{F}{\mathrm{DW}}_{\mathrm{AB}}T_{A}}{\times 10}^{-3}$$
where E_F_ is the exposure frequency (365 days/year); E_D_ is the exposure duration (70 years), equivalent to the average lifetime; F_IR_ is the food ingestion rate (345 g/person·day^−1^); C is the heavy metal concentration in plants (mg/kg); R_F_D is the daily reference dose (0.001 mg/kg·day^−1^); W_AB_ is the average body weight (70 kg); T_A_ is the average exposure time for non-carcinogens (365 days/year × ED); 10^−3^ is the conversion factor (mg/kg).

SPSS (version 25.0, Chicago, IL, USA) was used for the analysis of variance (ANOVA) of the data, and the LSD method was used for the analysis of significant differences (*p* < 0.05, *n* = 3).

## 3. Results

### 3.1. Effects of Cd Stress on Plant Biomass

After 40 days of liquid culture of *A. selengensis* under different concentrations of Cd stress, the growth rate (FW) of each treatment was retarded to different degrees compared with the CK (Figure 1a). The growth rate (FW) varied from 1.4–30.6% (Day 10) to 42.9–111.6% (Day 40) over time. A higher concentration of Cd treatment (25, 50, 100 μM) and longer time (Day 40) stress resulted in a significant decrease in the growth rate (FW) (*p* < 0.05). However, there was no significant difference in the growth rate of fresh weight under a lower concentration and short-term treatment, in comparison to the CK, i.e., CK (29.9%), and 1 μM (28.7%) at Day 10.

As shown in Figure 1b, low-Cd concentration (0.5 to 1 μM) had no significant effect on plant height, and plant height was significantly reduced when the Cd concentration was higher than 10 μM. As the stress concentration increased, the leaf size and number of leaves in the high stress concentration treatments (50 and 100 μM) decreased compared with the CK, and leaves showed chlorosis and stunting (Appendix A). It was worth noting that the main root length was shorter than that of the CK with the medium and high concentration treatments (10, 25, 50, and 100 μM); the number of lateral roots was also significantly reduced, and blackening symptoms appeared.

### 3.2. Effects of Cd Stress on MDA Concentration

As one of the most common biomarkers of lipid peroxidation, MDA was measured (Figure 2). Compared with the CK, Cd stress significantly increased the MDA content of *A. selengensis* over time (Figure 2). The content of MDA in the stem showed an overall increased from 2.78 to 7.30 nmol/g at Day 10. After 20–40 of days treatment, the content of MDA in leaves and stems apparently increased with the gradient Cd concentration, reaching the highest point at 53.03, 97.30, and 133.92 nmol/g (stem) and 23.04, 29.49, and 46.33 nmol/g (leaf) under 10 μM treatment, then it decreased slightly to 20.79, 61.56, and 106.29 nmol/g (stem) and 8.55, 17,80, and 33.95 nmol/g (leaf) under 100 μM treatment. Compared with the CK, all Cd treatments increased the content of MDA to varying degrees.

### 3.3. Distribution of Cd in Plant

The Cd content (FW) of roots, stems, and leaves is shown in Figure 3. The Cd accumulation in leaves, stems, and roots increased with gradient Cd concentrations as expected. However, the content of Cd in the three parts was significantly different in the order of roots > stems > leaves. The Cd content increased significantly when the stress concentration was higher than 10 μM. The Cd content in root was not significantly difference under the low-Cd stress treatments (0.5 to 1 μM). On the contrary, a significant difference appeared under the higher stress (25, 50, and 100 μM) treatments (*p* < 0.05). After 40 days of incubation, the Cd content in roots (0.67 to 30.95 mg/kg) was significantly higher than that in stems (0.37 to 8.13 mg/kg) and leaves (0.09 to 4.70 mg/kg).

### 3.4. TEM Analysis

In previous investigation, *A. selengensis* exhibited strong tolerance and accumulation capacity to metal ions. TEM was carried out to observe the morphology of the roots, stems, and leaves with and without the Cd treatment after 20 days (Figure 4). The results showed that there was no visible damage to the organelles in various parts of *A. selengensis* in CK (Figure 4a–c). The cytoplasmic membrane and cell wall surface were intact, smooth, and distinct, and the cytoplasmic membrane was close to the cell wall. The mitochondrial inner membrane was distinct and showed dense cristae; the chloroplast layer membrane structure was clear and complete, and arranged neatly. Compared to the CK, 10 μM Cd stress had a minor influence on the shape of the cellular structure. The root cell wall was thickened in the 10 μM treatment group (Figure 4d), but the damage was more serious under 100 μM treatment in that the cell wall had been broken into multiple parts. The 10 and 100 μM Cd stress caused damage to the mitochondrial cristae and vacuolization (Figure 4d,e,h), and the lamellar structure of the chloroplast was disordered and blurred (Figure 4f,i). The stem and leaf cells of the 100 μM treatment showed severe plasmolysis, and black deposits appeared on the cytoplasmic membrane and in the vacuole (Figure 4h,i).

### 3.5. Subcellular Distribution of Cd in Plant Roots

In order to further explore the distribution of Cd in cells, the distribution ratio of Cd in each part of the subcellular morphology of the *A selengensis* root was determined (Figure 5). As shown in Figure 5, more than half of the Cd accumulated on the cell wall, and the minority was distributed in organelles (mainly mitochondria and chloroplasts, etc.). The distribution of Cd in the cell wall (f1) accounted for 49–71% of the total Cd, while the soluble fractions (f3) accounted for 17–31% and the remaining 11–23% were organelles (f2). It can be seen that in the root, the proportion of Cd in f1 is significantly higher than that in f2 and f3.

### 3.6. FTIR Analysis

Our work found that the cell wall of the root might be the main site where Cd was adsorbed. To further characterize the adsorption sites, the FTIR analysis was performed on the root cell wall of CK, 10 and 100 μM (Figure 6). Compared with the CK, it was clear that the transmittance percentage of the peaks strengthened under the stress of metal. In addition, the characteristic peaks had shifted. Specifically, the peak at 3386 cm^−1^ (No. 1) was red-shifted by 11 and 15 cm^−1^ with the treatment of 10 and 100 μM Cd, and the peak at 2920 cm^−1^ (No. 2) had the same trend and were red-shifted by 2 and 5 cm^−1^, respectively. On the contrary, the peak at 1639 cm^−1^ (No. 3) was blue-shifted by 8 cm^−1^, and the stretching vibration peaks at 1056 cm^−1^ (No. 4) were blue-shifted by 11 and 15 cm^−1^.

### 3.7. Health Risk Assessments

TF and THQ were used to assess the health risk of *A. selengensis*. At the same stress concentration, the translocation factor (TF) of *A. selengensis* stems (Appendix A) decreased over time. After 40 days of treatment, the TF of stems and leaves were 0.25–0.40 and 0.12–0.23, respectively. Only in the 50 and 100 μM treatment, the leaf TF increased slightly over time, but it was still lower than that of other treatments. The TF of leaves was always lower than that of stems under the same treatment conditions. As shown in Table 1, continuous exposure to Cd resulted in a gradual increase in the THQ of stems and leaves. The THQ value of stems (2.00 to 44.52) and leaves (0.55 to 25.67) increased with the ascent of Cd concentration after 40 days, and the THQ value of stems was significantly higher than that of leaves under the same treatment.

## 4. Discussion

### 4.1. Effects of Cd Stress on Plant Growth

Cd is not an essential metal element for plants, and can lead to abnormal plant growth, such as chlorosis and shunned growth, which are common symptoms [24]. After long-term exposure to Cd, roots will exhibit necrosis, decompose and become viscous, thus reducing the elongation of roots and buds [25]. The toxicity of Cd will inhibit the formation of lateral roots, resulting in the hardening, distortion, and the browning of taproots [26]. In this experiment, the effect of Cd toxicity on root length was more significant than that on stem length (Appendix A), which significantly reduced the plant height of *A. selengensis* at concentrations higher than 10 μM (Figure 1b). This may be due to the high concentration of Cd stress inhibiting the mitosis of meristems, resulting in the reduction in root length [27]. This showed that *A. selengensis* had a certain degree of tolerance to lower concentrations of Cd stress or had a slight effect on growth. The content of Cd under 10 μM treatment is much higher than that in urban sewage, but the *A. selengensis* treated with less than 10 μM seemed to show a status with a slight inhibition of fresh weight and plant height. In soils with 1 and 10 mg/kg Cd stress, the length of the pakchoi’s root and stem were significantly inhibited compared with the CK [28]. A lower concentration of Cd stress had no effect on the fresh weight of roots and stems, while a higher concentration of Cd treatment significantly reduced the fresh weight of roots and stems by 23.36% and 29.58%, respectively [28]. After 15 days of parsley seedlings (*Petroselinum crispum* L.) treated with CdCl_2_ (75, 150, and 300 μM), the Cd content of the leaves was observed to be 0.52–2.57 mg/kg, but the plants showed no visible signs of poisoning [29]. This is similar to the phenomenon observed in this experiment. Under low-Cd stress (0.5 to 1 μM) and short-term (Day 10) growth conditions, there was no significant difference in the growth rate (FW) of *A. selengensis*, but the opposite was true at higher concentrations. The inhibition of Cd toxicity on biomass is more significant at high-Cd stress (25, 50 and 100 μM). Studies have also found that low Cd concentration could promote plant growth. After Cd stress was applied to 40 radish varieties, it was found that the biomass of 11 radishes grown in soil with a Cd content of 0.83 mg/kg for 53 days was significantly higher than that of the CK group (Cd content 0.31 mg/kg) and the biomass of 24 kinds of radishes did not significantly decrease compared with the CK [30].

### 4.2. Oxidative Stress Response of Plant to Cd Stress

Cd toxicity induces oxidative stress in plants, which is characterized by lipid peroxidation and the excessive production of oxygen free radicals, which leads to the damage of plant cell membranes, cell biomolecules and organelles [31,32]. ROS in plants such as hydrogen peroxide (H_2_O_2_) and malondialdehyde (MDA) is a major cause of damage to plants under stresses [33]. Malondialdehyde (MDA) is a product of membrane lipid peroxidation and can indirectly reflect the degree of membrane lipid peroxidation and cell membrane damage [34,35]. In this work, with the increase in the Cd concentration, the MDA content reached the maximum at 10 μM treatment, and then decreased. However, the content was always higher than the CK, indicating that the oxidation balance of *A. selengensis* was destroyed. The 6–24 h Cd stress treatment destroyed the oxidative balance of alfalfa, which was related to the rapid accumulation of peroxides and the decrease of homoglutathione and glutathione [36]. Another study found that the increase in MDA is related to the reduction in POD and SOD activity [37]. Under the high pressure of Cd, metal ions were absorbed by the plant for a long time and accumulated continuously, the concentration continued to increase to the point that the plant could not bear [38]. As a result, the metabolic activities of *A. selengensis* were affected, which caused the MDA content to decrease with the increase in Cd concentration (>10 μM). However, under low-Cd stress, due to the low Cd content in the plant, the toxic effect on the plant can be cured by its own regulation [39].

### 4.3. Roots Could Be the Main Part of Cd Enrichment

Studies have shown that the roots of plants have a strong fixation effect on Cd, which reduces the Cd content in the stems of some low-Cd cultivars [40,41,42]. Reducing the transport of Cd to the above-ground part is among the strategies for most leaf vegetables to deal with heavy metal poisoning [43,44]. Under normal circumstances, the roots of plants are the primary objects of direct contact with the soil, and Cd is absorbed by plants through the roots [45]. Only a part of Cd is transported to the above-ground parts of plants (i.e., stems, leaves, and reproductive parts), and follows the order of grains < fruits < leaves < roots [46]. In this experiment, the Cd content in the roots of *A. selengensis* was always higher than that in the stems and leaves, and the difference was significant. More than 57.8–70.8% (FW) of Cd was absorbed by the roots of *A. selengensis*, and the majority of the Cd was retained in the roots. Most Cd is fixed by underground organs and reduced transport to above-ground parts, which can alleviate the toxicity of Cd to plants [47]. A large number of studies have verified that the apoplastic and symplastic pathways are two adsorption pathways for plants to absorb metal ions such as Cd [48]. Metal ions flow in the apoplastic pathway in the free space. These free spaces are not restricted by the cell membrane, including metal ions passing through the cell wall and interstitial space [49]. The symplastic pathway is an active process that depends on metabolic activities. It is slower than the apoplastic pathway [48]. In soybeans (*Glycine max* L.), 98% of the total amount of Cd absorbed from the soil accumulates in the roots, and the rest is transferred to the above-ground part through vascular bundles [50]. However, there are exceptions. Some plant species, such as tobacco (*Nicotiana tabacum* L.), have higher metal accumulation potential [25]. Compared with roots, the Cd concentration accumulated in older tobacco leaves is higher than that of roots [25]. Therefore, the difference between species is an important factor when exploring the transport of Cd in plants.

### 4.4. Cd Stress Deformed the Subcellular Structure of Plants

Cd stress treatment caused varying degrees of damage to the ultrastructure of *A. selengensis* cells, and with the increase in the stress concentration, the damage to the cell structure became more serious, including the destruction of the cell membrane system and nucleic acid, photosynthetic protein damage, and the reduction in protein synthesis, which in turn affects the entire plant and prevents normal growth [26]. The primary sites of Cd harming the aerial part of the plant is the photosynthetic structure (mainly chloroplast) and the synthesis of photosynthetic pigments [51]. The excessive accumulation of Cd causes the deficiency of Mg, Ca, and Fe needed in the process of photosynthetic pigment synthesis [52]. This may be the cause of the chlorosis of *A. selengensis* leaves treated with a high concentration of Cd. Other studies have found that Cd stress causes the chlorosis of *A. thaliana* leaves, mainly due to the increase in the pectin and hemicellulose contents of the root cell wall under Cd stress, which leads to Fe being mainly retained in the roots, and hence the deficiency of Fe in the stems [53]. Compared with the stems and leaves, the cell wall structure of *A. selengensis* roots was more obviously damaged, which also verified the slow growth of *A. selengensis* at high concentrations. In the 10 μM treatment group, the thickening of the cell wall was observed, which was similar to the phenomenon of plants under Cd stress in other studies. In a study of tomato (*Phytolacca americana* L.) suspension cells, it was found that a 40% increase in the biomass of the cell wall was related to an increase in the content of Cd trace metals [54]. During the exposure of *Sorghum bicolor* to 100 μM CdCl_2_, the thickening of the cell wall appears in the xylem and phloem cells of the roots, and the same phenomenon occurs in the vascular bundle cells of the leaves [55]. This thickening of the cell wall is usually related to the increase in cellulose and lignin [56,57].

### 4.5. Root Cell Wall Might Play a Key Role in the Cd Accumulation

The cell wall is the first barrier for Cd to enter the protoplast [58], and it is the main location of Cd storage [59,60]. Subcellular analysis of the distribution of Cd in roots is of great significance for understanding the accumulation of Cd in *A. selengensis*. After the subcellular extraction of the roots of *A. selengensis*, it was found that the Cd content and the proportion of f1 were the highest. As the stress concentration increased, the proportion of the Cd content of f2 decreased. This may be due to the fact that plants reduce the transport of Cd to the organelles to alleviate the damage of Cd toxicity to the organelles. Wang et al. found that under low-concentration Cd stress, more than 50% of Cd is combined in the f1 of the root cells of pakchoi varieties Huajun2 and Hanlv, a large amount of Cd enters f3 under higher Cd stress, and the proportion of Cd in f1 is decreased [5]. Yang et al. found that the Cd distribution in rice roots is f3 > f1 > f2, and with the increase in the Cd stress, the proportion of Cd in the soluble and cell wall fraction increases significantly [61], which is similar to that in *Camellia sinensis* L. [62] and *Phytolacca americane* L. [63]. This may be due to the fixation of Cd in the cell wall after saturation; Cd is transported to the vacuole, and combined with the organic acid, organic base and protein in the vacuole to form a compartmentalization of the vacuole, thereby reducing the distribution of Cd in the organelles [63]. Vacuolar compartmentalization plays an important role in reducing free Cd in the cytosol [61]. Due to different experimental designs and plants, it is difficult to clarify the detoxification mechanism of Cd. However, it can be seen that cell wall fixation and vacuole compartmentalization are both Cd tolerance and detoxification mechanisms of plants.

The absorbance of the characteristic absorption peaks of each functional group of the root cell wall decreased compared with the CK, and the red-shift and blue-shift occurred to different degrees. This also showed that the molecular structure of the functional group on the cell wall changed after chelating Cd. After the root cell wall adsorbs Cd, it replaces part of the hydrogen in the -OH (Figure 6, No. 1), forming a stable compound, and causing the position of the -OH stretching vibration peak to move to a high wave number, which is mainly derived from the cell wall cellulose, hemicellulose, polysaccharides and other carbohydrates [64]. The characteristic peak at 2920 cm^−1^ (Figure 6, No. 2) has the same trend as the former, where it is mainly composed of protein, cellulose and pectin in the cell wall [64]. The pectin and hemicellulose on the cell wall contain various functional groups (e.g., -COOH, -OH, and -SH), which can bind metal ions [61], and the pectin fraction on the cell wall is considered to be the main metal binding site due to its high ion-exchange activity [58]. Cd will also induce the synthesis of proline-rich proteins, disease-related proteins, glycine-rich proteins, and other proteins to protect plants from heavy metal poisoning [65]. This is also the reason for the change in the characteristic peak of No. 3 in Figure 6 [66]. The stretching vibration peaks near 1054 cm^−1^ are mainly compounds such as alcohols, aliphatic groups, ether groups, and phenols [67,68]. Some studies have found that the content of phenols in the low-frequency region (1000 to 1200 cm^−1^) of plants increased after Cd stress [69]. The vicinity of the characteristic peak is a low-frequency region, the polysaccharide composition is relatively complex, and it is difficult to distinguish the specific functional group to which the characteristic peak belongs. The result showed that the carboxyl and hydroxyl groups of polar acidic compounds, cellulose, hemicellulose, and polysaccharides are the major Cd-binding sites.

### 4.6. The Health Risk Assessment

The TF and THQ related to the stems and leaves of *A. selengensis* are shown in Appendix A, Table 1. Since most of the Cd was absorbed and fixed by the roots, the TF of the stems and leaves decreased over time. According to the Integrated Risk Information System (IRIS) (US EPA, 2000), heavy metal has no significant effect on human health when the THQ < 1. In the actual planting process, the harvest cycle of edible *A. selengensis* is approximately 20 days, while the harvest time of *A. selengensis* used as medicinal materials is longer. In the experiment, *A. selengensis* on Day 20 had no health risk (THQ < 1) at low-Cd stress (0.5 to 1 μM). However, with the increase in the stress concentration and time, stems and leaves gradually showed obvious health risks and become increasingly serious. Hence, there are still health risks regarding the consumption of *A. selengensis* which is continuously exposed to Cd-contaminated soil. In addition, a report highlighted that the THQ of vegetables was higher in urban areas than that in rural areas, suggesting a higher potential health risk for urban residents and families with higher income levels [70].

## 5. Conclusions

In this experiment, it was found through liquid culture that a Cd stress concentration below 10 μM had no significant effect on the growth of *A. selengensis*. After 40 days, there was no significant difference between the plant height of the low-Cd treatment (0.5 to 1 μM) and the CK. This shows that *A. selengensis* has strong tolerance to low-Cd stress (0.5 to 1 μM). High-Cd stress and long-term inhibition (Day 40) seriously affected the normal growth of *A. selengensis*. The Cd content in roots is much greater than that in the above-ground part of plants, indicating that Cd is easily accumulated by the roots of *A. selengensis*. According to subcellular morphology observation, the root cell wall is the main site of Cd accumulation in roots, and the functional groups on the cell wall play a key role. In general, *A. selengensis* slows down the toxicity of Cd by root accumulation, the reduction in Cd transport, and cell wall fixation, which is consistent with the detoxification mechanism of most plants. The Cd transport factor (TF) was less than one, which further certifies that *A. selengensis* is a root-accumulating plant. The health risk assessment showed health risks in the aerial parts of *A. selengensis* under high-concentration Cd pressure and long-term treatment (THQ > 1). Therefore, to ensure the safe consumption of *A. selengensis* in field cultivation, metal passivators, soil conditioners, or agronomic regulation can be used to improve the rhizosphere soil environment and reduce the accumulation of Cd. 

## Figures and Tables

**Figure 1 ijerph-19-01183-f001:**
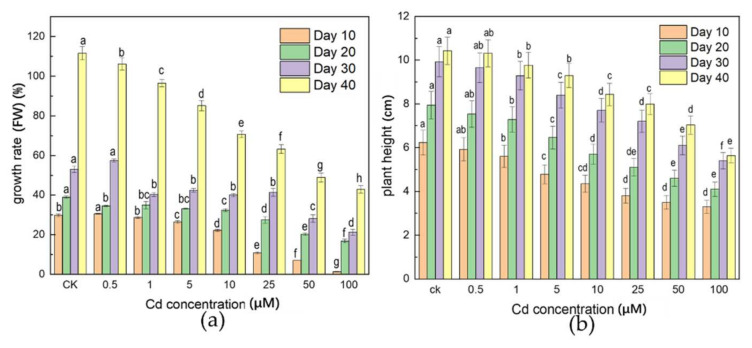
The effect of different concentrations of Cd on the growth rate (FW) and plant height of *A. selengensis*. (**a**) growth rate (FW), (**b**) plant height. The results are means ± SD (*n* = 3), lower letters mean significant differences (*p* < 0.05) between different Cd concentrations at the same treatment time.

**Figure 2 ijerph-19-01183-f002:**
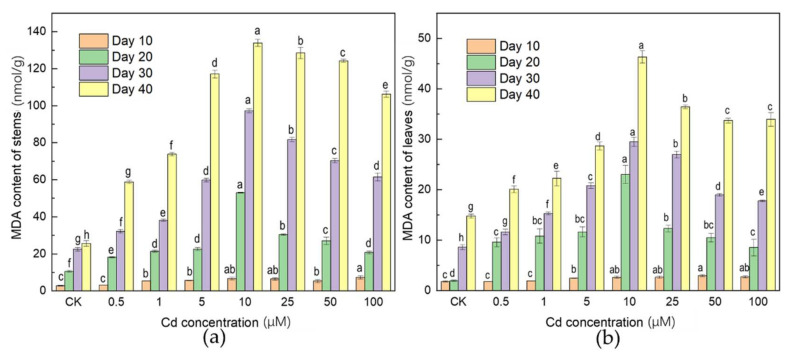
Changes of MDA content of *A. selengensis* after treatment with different concentrations of Cd, (**a**) stem, (**b**) leaf. The results are means ± SD (*n* = 3), lower letters mean significant differences (*p* < 0.05) between different Cd concentrations at the same treatment time.

**Figure 3 ijerph-19-01183-f003:**
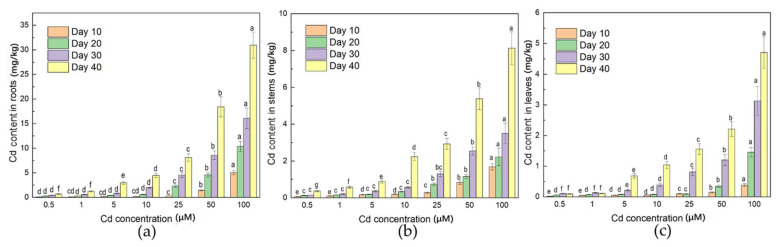
Fresh weight Cd content of *A. selengensis* after treatment with different concentrations of Cd, (**a**) root, (**b**) stem, (**c**) leaf. The results are means ± SD (*n* = 3), lower letters mean significant differences (*p* < 0.05) between different Cd concentrations at the same treatment time.

**Figure 4 ijerph-19-01183-f004:**
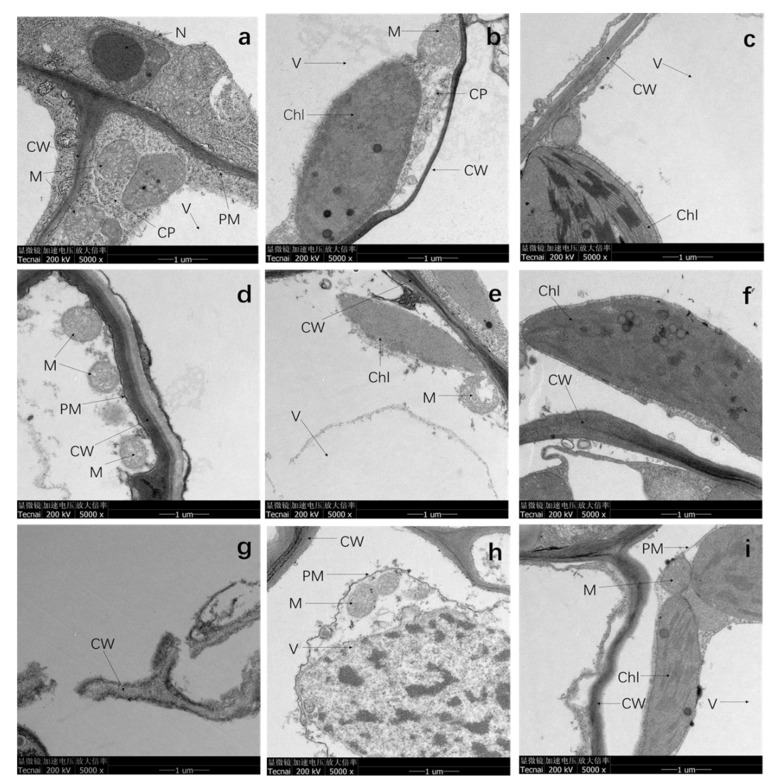
(**a**–**c**) are the root, stem, and leaf of the CK group, (**d**–**f**) are the root, stem, and leaf of the 10 μM treatment group, and (**g**–**i**) are the root, stem, and leaf of the 100 μM treatment group. Chl—chloroplast; CP—cytoplasm; V—vacuole; CW—cell wall; M—mitochondria; PM—cytoplasmic membrane; N—nucleus.

**Figure 5 ijerph-19-01183-f005:**
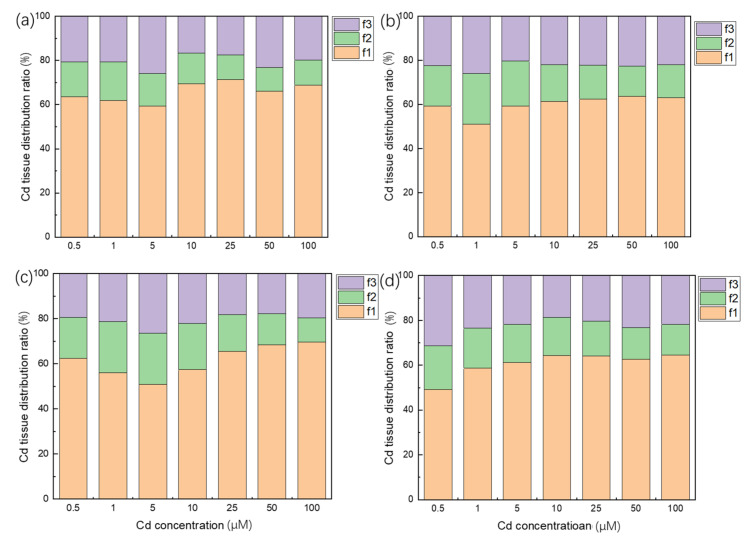
Distribution ratio of Cd content of each fraction of subcellular, f1: cell wall fraction, f2: organelle fraction, f3: soluble fraction, under different Cd concentration treatments (**a**) hydroponic culture for 10 days, (**b**) hydroponic culture for 20 days, (**c**) hydroponic culture for 30 days, (**d**) hydroponic culture for 40 days.

**Figure 6 ijerph-19-01183-f006:**
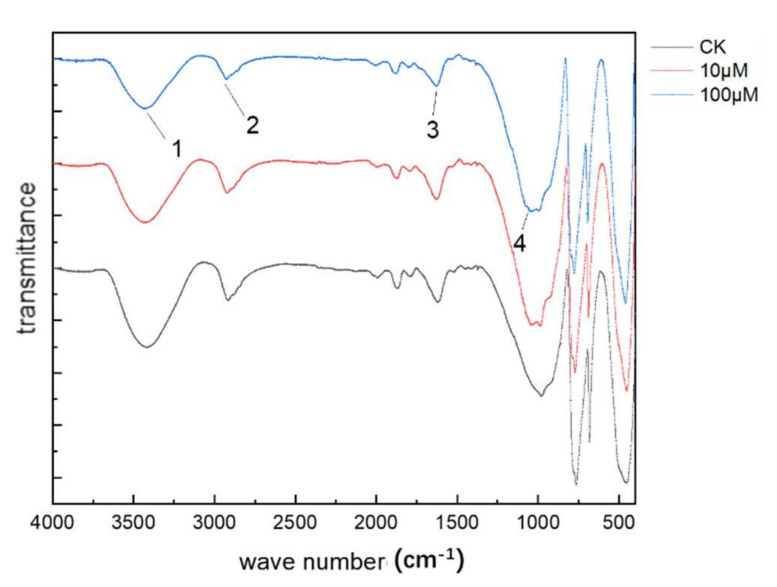
The FTIR spectra of cell wall components of roots after 40 days of cultivation.

**Table 1 ijerph-19-01183-t001:** Target hazard quotient (THQ) values of *A. selengensis* after 40 days of liquid culture.

Cd	Day 10		Day 20		Day 30		Day 40	
Concentration (μM)	Leaf	Stem	Leaf	Stem	Leaf	Stem	Leaf	Stem
0.5	0.07	0.31	0.30	0.68	0.40	0.80	0.55	2.00
1	0.17	0.50	0.37	0.81	0.73	1.07	0.64	3.17
5	0.30	0.96	0.45	1.06	1.21	2.25	3.77	4.90
10	0.32	1.12	0.44	1.86	2.13	3.04	5.65	12.15
25	0.57	1.55	0.90	4.78	4.46	7.64	8.62	16.00
50	0.79	4.71	1.89	6.32	5.47	10.10	12.47	29.38
100	2.23	9.04	8.01	14.72	17.26	26.71	25.67	44.52

## Data Availability

The data presented in this study are available on request from the corresponding author.

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
