# Peer review of "Accumulation Mechanism and Risk Assessment of Artemisia selengensis Seedling In Vitro with the Hydroponic Culture under Cadmium Pressure"

_ijerph, 2022, doi:10.3390/ijerph19031183_

Round 1
Reviewer 1 Report
I have added my comments to the attached file.

Reviewer 2 Report
The present manuscript describes the accumulation of Cd in a medicinal edible plant.
The in vitro study is well presented and coherent. However, from my point of view the manuscript needs a deeper chemical research in order to enhance his scientific soundness and understand better the effects of the Cd in the plant. A phytochemical analysis of primary and secondary metabolites will be necessary.
Author Response
Response to Reviewer 2 Comments
Point 1: The in vitro study is well presented and coherent. However, from my point of view the manuscript needs a deeper chemical research in order to enhance his scientific soundness and understand better the effects of the Cd in the plant. A phytochemical analysis of primary and secondary metabolites will be necessary.
Response 1: We would like to thank you for your positive and encouraging comments on our manuscript. We refer to some studies on oxidative stress and add relevant analysis to the manuscript.
ROS in plants such as hydrogen peroxide (H2O2) and malondialdehyde (MDA) is a major cause of damage to plants under stresses (line 336-338).
The 6-24h Cd stress treatment destroyed the oxidative balance of alfalfa, which was relat-ed to the rapid accumulation of peroxides and the decrease of homoglutathione and gluta-thione. Another study found that the increase in MDA is related to the reduction in POD and SOD activity (343-346).
Round 2
Reviewer 1 Report
Thank the authors for providing changes, English corrections and responses to my questions.
The text has been impoved significantly. However, there are still some issues to be corrected. I have listed some of them below. I recommend the authors to check the text thoroughly word by word to unify the corrections.
L 17 – compositae (Compositae)
L 18 and L 34 – cadmiun (spelling)
L 47 entery into humans (entry?)
L 82 – in vitro (italics)
L 90 – couldprovide (space between words)
Besides, there are many places were space between word and unit in bracket is missing – in figures (axis description)
Chapter 2.2 – values and units are written without a space between them (such as 3ml instead of 3 ml).
L 119 – the same (2ml)
Is there a reason for writting 0.5000 g (L 109) or 1.000 g (L 126) vs. 0.5 g (L 116?) – I suppose that the number of digits could be unified.
L 138 1 ~ 2 (does it mean 1 or 2 hours? Or „one to two hours“? It the second case, I would prefer 1-2. Similar problem in previous text (abstract and Introduction, e.g. L 66) – I am not sure it the ~ symbol is correct.
L 179 – another example (1.4% ~ 30.6%) – the symbol ~ is usually used as „around“. Here, I would recommend to use 1.4% – 30.6%.
These cases are many times within the manuscript.
However, there are places where the symbols or words are used more clearly, such as:
L 201-202: „The content of MDA in the stem showed an overall increased from 2.78 to 7.30 nmol/g at Day 10. After 20-40 of days treatment,“.
L 183 short-termtreatment (space between words)
L 217 The Cd content was not significantly difference under the low-Cd stress treatments (0.5 to 1 μM). - The sentence need a correction. (was not significantly different?)
L 239 blurred(Figure 4f, i). – space between words
L 262 mightbe (the same problem)
L 398 Camellia Sinensis L. (Camellia sinensis L.)
L 445 aCd (space)
